# Biological Actions of Alamandine: A Scoping Review

**DOI:** 10.3390/biomedicines13081957

**Published:** 2025-08-11

**Authors:** Juliane Flor, Andresa Thomé Silveira, Isabel Amaral Martins, Laura Bastos Otero, Flávia Moraes Silva, Adriana Fernanda K. Vizuete, Márcia Rosângela Wink, Katya Rigatto

**Affiliations:** 1Laboratório de Fisiologia Translacional, Pós-Graduação em Ciências da Saúde, Universidade Federal de Ciências da Saúde de Porto Alegre—UFCSPA, Porto Alegre 90050-170, RS, Brazil; juliane.flor@ufcspa.edu.br (J.F.); andresasilveira@ufcspa.edu.br (A.T.S.); isabel.martins@ufcspa.edu.br (I.A.M.); lauraos@ufcspa.edu.br (L.B.O.); adriana.vizuete@ufcspa.edu.br (A.F.K.V.); 2Pós-Graduação em Ciências da Saúde, Universidade Federal de Ciências da Saúde de Porto Alegre—UFCSPA, Porto Alegre 90050-170, RS, Brazil; flaviams@ufcspa.edu.br (F.M.S.); mwink@ufcspa.edu.br (M.R.W.); 3Departamento de Nutrição, Universidade Federal de Ciências da Saúde de Porto Alegre—UFCSPA, Porto Alegre 90050-170, RS, Brazil; 4Departamento de Ciências Básicas da Saúde, Universidade Federal de Ciências da Saúde de Porto Alegre—UFCSPA, Porto Alegre 90050-170, RS, Brazil; 5Laboratório de Biologia Celular, Universidade Federal de Ciências da Saúde de Porto Alegre—UFCSPA, Porto Alegre 90050-170, RS, Brazil

**Keywords:** Alamandine, mechanism of action, therapeutic potential, AMPK, nitric oxide, MAPK

## Abstract

**Objective**: This scoping review aims to comprehensively map the existing literature on the mechanisms of action of Alamandine (ALA), a peptide within the renin–angiotensin system, and its effects across various physiological systems. **Materials and Methods**: Utilizing the Joanna Briggs Institute methodology, a thorough search of databases including PubMed, Embase, Scopus, and Web of Science was conducted up to 30 January 2025. The review focused on identifying studies that explore the biological and therapeutic roles of ALA in different contexts, incorporating in vivo, in vitro, and in silico research. **Results**: A total of 590 records were initially identified, with 25 meeting the eligibility criteria for inclusion in this review. China emerged as the leading contributor to the research in this area, with a significant focus on the cardiovascular system. The studies revealed that ALA exhibits a range of beneficial effects, including anti-inflammatory, vasodilatory, antifibrotic, and antiapoptotic actions. These effects are primarily mediated through the inhibition of the mitogen-activated protein kinase (MAPK) signaling pathway and modulation of the nitric oxide pathway. The review also highlighted AL’s potential in mitigating oxidative stress and its implications in treating cardiovascular diseases, fibrosis, and cancer. **Conclusions**: The findings suggest that ALA holds significant therapeutic potential, offering antihypertensive, anti-inflammatory, antifibrotic, and anticancer benefits without notable adverse effects, warranting further research to explore its full potential and mechanism of action.

## 1. Introduction

The classical renin–angiotensin system (RAS) has long been recognized as a fundamental regulator of blood pressure, metabolism, fluid and electrolyte balance, and cardiovascular and renal homeostasis. It is activated in response to decreases in blood volume or pressure, beginning with the release of renin by the kidneys. Renin catalyzes the conversion of angiotensinogen, which is subsequently converted into angiotensin II (Ang II) by the angiotensin-converting enzyme (ACE)—the primary effector peptide of this axis. Ang II exerts potent vasoconstrictive, prohypertensive, and sodium-retentive effects through the activation of the angiotensin type 1 receptor [1]. These actions include increased sympathetic tone, aldosterone secretion, and fluid retention, all of which are essential for short-term regulation of blood pressure and volume homeostasis [2].

At the molecular level, Ang II contributes to tissue damage through several interconnected mechanisms. It induces oxidative stress via stimulation of NADPH oxidase, promotes inflammation and fibrosis through activation of intracellular signaling cascades, and modulates key pathways such as Nuclear Factor Kappa B (NF-κB), phosphoinositide 3-kinase (PI3K)/Protein Kinase B (Akt), and various mitogen-activated protein kinases (MAPKs), including extracellular signal-regulated kinases (ERKs) 1/2, c-Jun N-terminal kinase, and p38. The downstream consequences—such as endothelial dysfunction, cellular hypertrophy, and apoptosis—will be discussed in detail throughout this review, along with the corresponding literature references.

Building upon this evolving view of the RAS, ALA was identified during an investigation into the function of angiotensin-converting enzyme 2 (ACE2), a component of the RAS [3]. This discovery, alongside the identification of Angiotensin-(1-7) [Ang-(1-7)] in 1988 [4], marks a significant milestone in the interpretation of the RAS.

The classical axis coexists with a novel axis involving Ang-(1-7)-G protein-coupled receptor (Mas)/ALA-Mas-related receptor member D (MrgD), which plays a central role in the pathophysiology of various diseases. The primary mechanism by which ALA functions is through the activation of MrgD. This activation leads to various downstream effects, including vasodilation, anti-inflammatory actions [5], and antifibrotic effects [6], which closely resemble those of Ang-(1-7) [7].

Literature data demonstrate that ALA induces vasodilation both in vitro and in vivo, likely through the activation of nitric oxide synthase (NOS) to produce nitric oxide (NO) [8]. Furthermore, ALA exhibits antifibrotic effects, likely through the inhibition of transforming growth factor-β (TGF-β) signaling, thus reducing collagen deposition in the heart [9] and kidneys [10].

Building on this understanding, we propose a scoping review using an emerging methodology to synthesize existing knowledge on the topic. Although not all articles present the highest quality, considering the limited literature available, all studies discussing ALA0’s mechanisms of action were included. This method also identifies gaps, opportunities, and research priorities to guide future studies on its mechanisms of action in the body.

## 2. Materials and Methods

This scoping review employs the Joanna Briggs Institute methodology [11] and is registered on the Open Science Framework (OSF, DOI 10.17605/OSF.IO/CG9U4). The review follows five stages: identifying research questions, finding relevant studies, selecting eligible studies, collecting and recording data, and summarizing and reporting results. Detailed descriptions of each stage are provided below [11].

### 2.1. Stage 1: Identification of the Research Question

The main research question was formulated using the ‘Population’, ‘Concept’, and ‘Context’ (PCC) framework, covering both in vivo and in vitro studies. This approach focuses on identifying the primary signaling pathways that explain the ALA’s mechanism of action. In this context, we considered all body systems, including the kidneys, brain, cardiovascular system, heart, eyes, lungs, and others.

Based on the PCC framework, our main research question was the following: What mechanisms of action of ALA are described in the literature? This led to the following sub-questions:In what clinical conditions and outcomes has ALA been described?What are the effects of ALA?

### 2.2. Stage 2: Identification of Relevant Studies (Search Strategy)

All studies that evaluated the mechanism of action of ALA, regardless of the body system studied, were eligible for this scoping review. We did not impose restrictions on publication date or language. However, qualitative studies, reviews, editorials, comments, abstracts, and conference proceedings were excluded.

Initially, a search was conducted in the PubMed database to identify potential keywords for the search strategy, based on the titles, abstracts, and keywords of retrieved articles. Subsequently, a final search was performed across the PubMed, Embase, Scopus, and Web of Science databases (see Appendix A for all search strategies). All databases were searched on 14 April 2023 and updated on 30 January 2025.

### 2.3. Stage 3: Selection of Studies for the Review

Endnote^®^ (X19) was employed as reference management software to facilitate data management. Duplicate studies were removed using the software’s automated deduplication function. Subsequently, Rayyan^®^ (https://www.rayyan.ai/) software was utilized, and two independent reviewers (A.T.S. and J.F.) assessed the eligibility of each report through a two-step process. Initially, they reviewed the titles and abstracts to select all potentially eligible articles. Later, A.T.S. and J.F. read the full texts to confirm eligibility. Any discrepancies between the reviewers were resolved through consensus, and if necessary, a third reviewer (I.A.M.) was consulted.

### 2.4. Stage 4: Data Extraction

The same two reviewers (A.T.S. and J.F.) independently extracted data from all eligible studies using a data extraction spreadsheet created in Microsoft Office Excel^®^. A pilot test of this spreadsheet was conducted using five randomly selected full texts before proceeding with data extraction, ensuring consistent data extraction by the reviewers and avoiding ambiguities and errors. Appendix A summarizes the extracted data.

### 2.5. Stage 5: Summary of Data and Synthesis of Results

We used descriptive statistics to outline the characteristics of studies on ALA mechanisms of action, incorporating tables and graphs to present the collected data. This scoping review is presented in accordance with the Preferred Reporting Items for Systematic Reviews and Meta-Analyses for Scoping Reviews (PRISMAScR) checklist (Appendix A) [12].

## 3. Results

### Selection of Studies and General Characteristics of Selected Studies

Figure 1 presents the study selection flowchart. A total of 590 records were identified across all databases. After automated deduplication and screening of titles and abstracts, 67 records remained for full-text examination and data extraction. Of these, 42 studies were excluded for the following reasons: 27 did not address the research question, 13 were posters, and 2 were abstracts. Consequently, 25 studies were deemed eligible [6,8,9,13,14,15,16,17,18,19,20,21,22,23,24,25,26,27,28,29,30,31,32,33,34]. Appendix A provides a list of excluded studies along with justifications. Detailed characteristics of each study are presented in Appendix A. Most studies were published in 2022 (n = 6) [9,13,15,16,17,18].

Figure 2 provides a summary of the study features, including publication year, country, impact factor, and body system. All studies were described in English and were experimental in nature. Most of the studies were published in journals with an impact factor above 4.5, with the cardiovascular system being the most frequently investigated.

Geographically, most of the research is emerging from China and Brazil, indicating a strong local interest. This may reflect strategic research priorities or funding opportunities in these regions. Other countries have contributed fewer studies, suggesting a more limited yet global interest in the topic.

Temporally, there is an observable increase in publications from the late 2010s to the mid-2020s, peaking in the early 2020s, which coincides with growing recognition of ALA’s physiological roles. The predominance of cardiovascular system studies aligns with the molecule’s established relevance in this area, but research has also expanded into renal, metabolic, and other biological systems, illustrating its multifaceted physiological impact.

These trends reflect an expanding and geographically diverse research landscape on ALA, highlighting its potential importance across multiple systems and justifying further investigation in both established and emerging regions of interest.

Figure 3 illustrates the timeline of significant discoveries related to ALA’s mechanisms of action since its identification in 2013. This figure highlights the evolving focus of research, particularly emphasizing its role within the cardiovascular system. Initially, research centered on ALA’s potential to enhance heart function and manage hypertension.

From 2018 onwards, studies expanded to explore its anti-inflammatory and antifibrotic properties, with particular attention to its interactions with signaling pathways like MAPK and Adenosine Monophosphate-activated protein kinase (AMPK). These discoveries underscore ALA’s therapeutic promise in addressing oxidative stress and fibrotic diseases. This timeline underscores the ongoing importance of ALA in biomedical research, highlighting the need for further studies to explore its therapeutic applications in areas such as metabolic disorders or neuroprotection.

## 4. Discussion

Identified in 2013 [35], ALA can be derived from angiotensin A through the action of ACE2 [36] or by the decarboxylation of Ang-(1-7) [37] (Figure 4).

Activation of the ALA-MrgD axis is associated with vasodilation, anti-inflammatory, and antifibrotic actions. These effects are mediated by signaling pathways (Figure 5), particularly mitogen-activated protein kinases (MAPKs) and AMPK, regulating NO production and cellular energy homeostasis.

### 4.1. Mas and MrgD Receptors: Alamandine’s (Un)Specificity

The MrgD and Mas receptors are expressed in the dorsal root ganglia of the nervous system [38] and various organs, including the heart [8,39], brain [40,41,42], lungs [22], adipose tissue [43], vascular endothelium, arterial smooth muscle cells [44], and retina [33]. ALA’s affinity for the Mas receptor indicates a secondary receptor role [16,37].

The Mas and MrgD receptors may interact to form a functional complex. ALA binds to MrgD, promoting the dimerization of Mas and MrgD receptors, which leads to anti-inflammatory responses like reduced interleukin-6 and interleukin-1β secretion in lipopolysaccharide-activated THP-1 macrophages and M1 macrophages. Moreover, ALA promotes antiproliferative effects [45].

The complexity of this interaction is evident in cells using compounds such as PD123319, D-PRO7-ANG-(1-7), and A779, exposing the challenge of targeting specific receptors [46]. This suggests ALA’s actions may involve MrgD and AT2 receptors, not just the Mas receptor [3,32,42].

### 4.2. MAPK and AMPK Signaling Pathways

ALA-MrgD interaction in cardiomyocyte cultures enhances NO production [8] and phosphorylates AMPK and LKB1, key regulators of energy and cardioprotection [47]. AMPK activation is crucial for preventing cardiac injury progression to heart failure [48,49] and is linked to metabolic diseases [50,51], with bone tissue research supporting these findings [23].

Emphasizing its cardiovascular benefits, ALA exhibits a broad anti-inflammatory effect by reversing the increased phosphorylation of PI3K, Akt, ERK, c-Jun N-terminal kinase, and p38, reversing their increased activity in sepsis-associated renal injury [20]. This inhibition involves the protein kinase A (PKA) signaling pathway [52], underscoring cyclic adenosine monophosphate-dependent kinase’s role in these processes [53]. Inhibition of PKA by ALA induces cardiac remodeling [52] and promotes vasodilation, as confirmed using KT5720, a PKA inhibitor [25]. In fact, Shen et al. (2018) [25] demonstrated that regulation of the PKA pathway is essential for ALA’s effects in the brain.

Additionally, in ischemia–reperfusion, ALA treatment significantly lowers apoptosis in myocardial cells [27], possibly by attenuating the inflammatory response through inhibition of NF-κB activation [27,33]. This suppression reduces pro-inflammatory cytokine expression [20,54] and reactive oxygen species production [33] and impacts myocardial infarction models due to pressure overload [9]. It promotes protein kinase phosphorylation [20,55], affecting the balance between cellular survival and apoptosis.

The anti-inflammatory properties of ALA also extend to synovial fibroblasts, inhibiting the MAPK pathway and leading to a reduction in inflammatory mediators associated with rheumatoid arthritis [15]. Furthermore, ALA decreases matrix metalloproteinase-2, essential for TNF-alpha maturation and apoptosis control in cardiomyocytes [55]. It also reduces TGF-β, preventing connective tissue fibrosis, and plays a crucial role in mitigating hypertrophic and fibrotic pathways by inhibiting phosphorylated forms in the MAPK pathways [6].

Conversely, Ang II is known to promote phosphorylation of p38 MAPK, contributing to cardiac fibrosis [56], vascular fibrosis [32,57], cardiac stress, and hypertrophy [58,59]. The Ang-(1-7) and ALA axis improves aortic function [60], suggesting therapeutic benefits.

ALA also improves cardiac function in models of heart failure by attenuating TGF-β signaling and ERK 1 and 2 (ERK1/2) phosphorylation [26]. In cardiomyocytes, PKA inhibition may reduce MAPK activation, affecting pathways like c-Jun N-terminal kinase and p38 [6,53,61].

### 4.3. Nitric Oxide Production: The Therapeutic Role of ALA in Cardiovascular Health

ALA enhances endothelial nitric oxide synthase activity at serine 1177 and threonine 495, increasing NO release and highlighting its therapeutic potential for vasodilation and cardiovascular health [24,25,52]. Studies on goldfish (*Carassius auratus*) show that ALA enhances cardiac contractility under normoxic conditions via the NOS/NO system, revealing its sensitivity to hypoxia and broad cardiac benefits [16,62].

Furthermore, Songür et al. (2023) [28] and Hu et al. (2021) [20] showed that ALA’s anti-inflammatory, antiapoptotic, and antipyretic properties protect against acute renal injury and endotoxemia by suppressing inducible NOS (iNOS) expression, highlighting its potential in mitigating inflammatory responses. In adipose tissue, ALA activates the MrgD receptor, triggering c-Src and increasing iNOS expression [29], leading to NO production that may cause mitochondrial dysfunction and stimulate lipoprotein lipase activity, and adversely affecting adipocytes [63].

In a transgenic rat model with overexpressed renin, ALA enhances cardiomyocyte contractility via the NO/calcium/calmodulin-dependent kinase II (CaMKII) pathway, initiating NO production and activating CaMKII in vascular smooth muscle cells [21]. CaMKII phosphorylates the threonine 17 residue of phospholamban within the sarcoplasmic reticulum, increasing the activity of the sarcoplasmic reticulum Ca^2+^-ATPase and enhancing contractility [21,64].

### 4.4. Modulation of PI3K Enzyme Activity by ALA

The PI3K enzyme family is central to numerous biological processes, including cell survival, apoptosis, and cardiac function [65], playing a significant role in cardiac hypertrophy and diastolic dysfunction associated with hypertension. In primary cardiac fibroblasts, ALA’s interaction with the MrgD receptor inhibits Akt activation induced by Ang II, thereby reducing cardiac fibrosis [30,66,67].

PI3K’s role in cellular processes is broad. The inhibition of the PI3K/Akt pathway is also relevant in cancer cell lines such as Mia Paca-2 and A549. The ALA treatment shifts energy generation from anaerobic to aerobic processes, potentially slowing cancer growth. It inhibits the PI3K/Akt/mTOR pathway, essential for cancer cell growth and survival, and induces the nuclear translocation of the FoxO1 protein, affecting various cellular processes [13].

This pathway inhibition leads to the dephosphorylation and reduced activity of components like 3-phosphoinositide-dependent kinase 1, Akt1, and the mTOR receptor, as well as the BRAF/MKK/ERK1/2 signaling pathway, resulting in cell cycle arrest and apoptosis in pancreatic cancer cells. ALA reduces the phosphorylation of ERK1/2 and its effectors, NIBAN2 and Stathmin 1, highlighting its potential as a therapeutic target in cancer treatment [13,68].

Moreover, Wang et al. (2023) [31] reported that ALA inhibits glycolysis in vitro, via the MrgD axis, and downregulates hexokinase 2 (HK2), reducing the impact of TGF-β1 on lung fibroblasts [31]. ALA decreases the NADPH Oxidase 4 levels and reactive oxygen species generation in fibroblasts, which are known to contribute to the development of pulmonary fibrosis [22]. Under anaerobic conditions, the glycolysis enhancement results in lactate formation from pyruvate [69], creating a feedback loop that enhances lung fibroblast activation and upregulates glycolysis pathways such as HK2 activity and the allosteric effector of phosphofructokinase-1 by 6-phosphofructo-2-kinase/fructose-2,6-bisphosphatase 3 (PFKFB3) activity. In fact, HK2 and PFKFB3 are related to fibrosis development in several tissues: pulmonary [70,71], hepatic stellate cells [72,73], cardiac [74], kidney tubular cells [75], and renal fibroblast NRK-49F cells [76].

Conversely, ALA/MrgD counter-regulates this loop, reducing hexokinase activity [77] and decreasing glycolysis pathways [31]. Moreover, previous studies suggest that ALA-MrgD signaling downregulates the expression of HK2 and PFKFB3 genes, reinforcing the importance of changes in energetic metabolism during fibrosis and highlighting ALA’s potential role in controlling this energetic shift [31,73,75,76,78].

Mitophagy and glycolytic flux indicate a complex interaction with ALA [79]. These insights highlight ALA’s therapeutic potential by inhibiting lung fibroblast activation via TGF-β1 and autophagy proteins (Parkin/LC3) while counteracting metabolic reprogramming in bleomycin-induced fibrosis by suppressing HK2, PFKFB3, Parkin, and LC3 activation [31].

These findings show that in adipose tissue, ALA activates the NFκB and p38 MAP kinase pathways, which upregulate plasminogen activator inhibitor-1, a protein linked to adverse health outcomes [80]. Uchiyama et al. (2017) [29] demonstrated that a low dose of ALA (5.76 μg/kg) modulates leptin expression and secretion via the phospholipase C, cSrc, and p38 MAPK pathways. These findings reveal ALA’s intricate metabolic effects, indicating the need for further investigation to fully understand its mechanisms and potential applications.

### 4.5. Alamandine and Oxidative Stress

Evidence suggests that ALA plays a crucial role in reducing cardiac fibrosis resulting from oxygen and glucose deprivation. This effect is supported by reductions in oxidative stress and ischemic injury [9], as well as a decrease in apoptosis [9,81], which collectively lowers the risk of heart failure [82]. ALA administration is associated with enhanced activity of superoxide dismutase (SOD), catalase, and glutathione peroxidase, as demonstrated in the hippocampus of C57/Bl6 mice [17]. Furthermore, De Lima Sanches et al. (2023) [14] confirmed ALA’s antioxidant properties, showing that oral administration effectively reduced superoxide anion (O_2_^•−^) and normalized nuclear factor erythroid 2-related factor 2 levels in the right carotid artery following transverse aortic constriction.

Oxidative stress has been related to caspase-3 signaling and cardiac apoptosis activation, particularly in the doxorubicin (DOX) model [83]. Hekmat et al. (2021) [19] demonstrated increased caspase-3 activation and apoptosis in DOX-exposed groups versus controls, while ALA significantly reduces both. Zhao et al. (2022) [9] also demonstrated that ALA inhibits increases in collagen I, α-SMA, TGF-β, Bax/Bcl2, and the caspase-3/cleaved caspase-3 ratio, processes linked to cell death from oxygen–glucose deprivation in neonatal rat cardiac fibroblasts, highlighting ALA’s cardiac protective effects [9].

According to J. Zhu et al. (2021) [34], in rats subjected to ischemia–reperfusion injury, ALA treatment significantly increased SOD activity while reducing levels of malondialdehyde, NADPH oxidase activity, O_2_^•−^ production, Nox expression, inflammation, and apoptosis. Additionally, Gong and colleagues [18] found that in cases of renal sodium overload, ALA mitigated the decline in SOD and glutathione peroxidase activity, as well as the increase in malondialdehyde, 8-hydroxy-2-deoxyguanosine, and O_2_^•−^ levels. This effect was likely mediated by inhibiting the protein kinase C signaling pathway [18].

Despite the growing interest in ALA and its promising biological effects, important knowledge gaps remain. One major limitation is the lack of comprehensive studies examining its receptor specificity, as ALA appears to interact not only with the MrgD receptor but also potentially with Mas and AT2 receptors. Moreover, most evidence comes from experimental models using varying doses, administration routes, and disease contexts, which complicates comparisons across studies. The tissue-specific effects of ALA, as well as its dual roles, are protective in some systems but potentially detrimental in others (e.g., adipose tissue). Another limitation is the lack of clinical research, with no human studies to date assessing the safety, pharmacokinetics, or therapeutic efficacy of ALA.

## 5. Conclusions

Despite our limited knowledge, the discovery of Ang-(1-7) and ALA represents a significant advancement in our understanding of the RAS by introducing peptides with antagonistic actions. The recognition of these new peptides has sparked a debate and reinterpretation of the RAS. Importantly, studies indicate that ALA administration, across a wide range of dosages, does not produce detrimental effects.

It is crucial to recognize that ALA’s discovery is recent, and many mechanisms require further elucidation. Our current understanding of its role in pathophysiology and its mechanisms of action is based on the findings presented in this article. Moreover, these findings emphasize the need to consider dosage, tissue specificity, and the experimental conditions (in vivo vs. in vitro) when evaluating ALA’s biological impacts. Therefore, additional studies are necessary to explore ALA’s diverse roles in health and disease across various pathophysiological contexts.

Future research should prioritize the standardization of experimental protocols to facilitate better comparison of results and clarify the specific pathways involved in ALA signaling. These promising findings justify further preclinical studies and pave the way for translational efforts that may ultimately support clinical application.

## Figures and Tables

**Figure 1 biomedicines-13-01957-f001:**
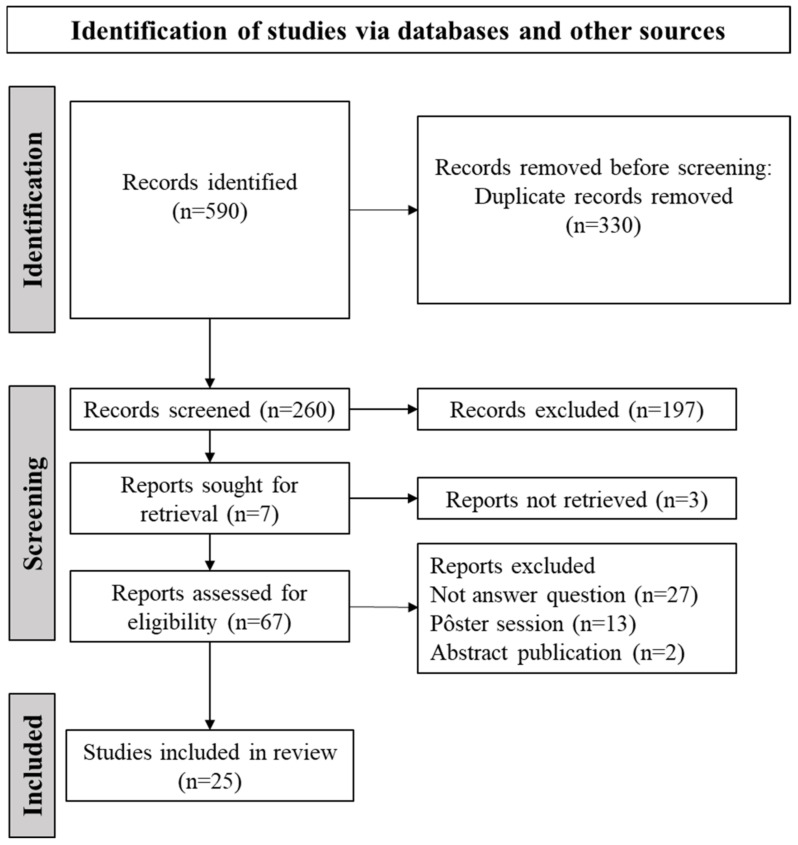
Visual representation of study selection for the current scoping review.

**Figure 2 biomedicines-13-01957-f002:**
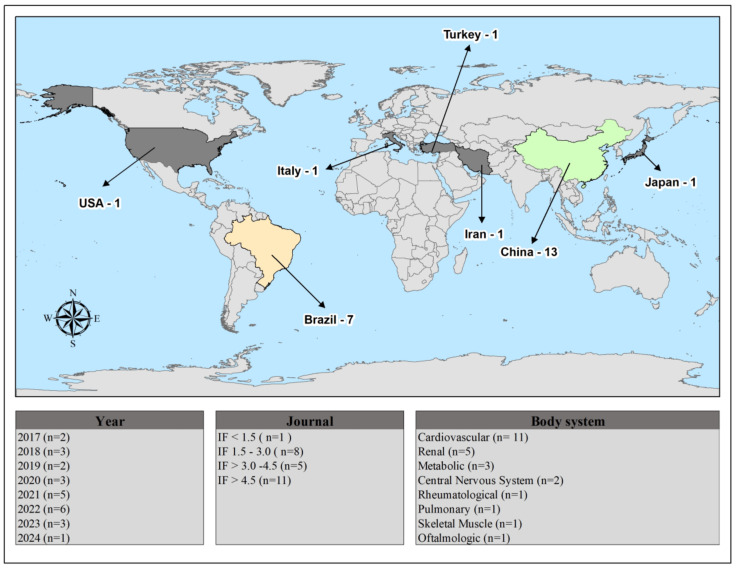
Geographic and scientific distribution of original research articles included in this review.

**Figure 3 biomedicines-13-01957-f003:**
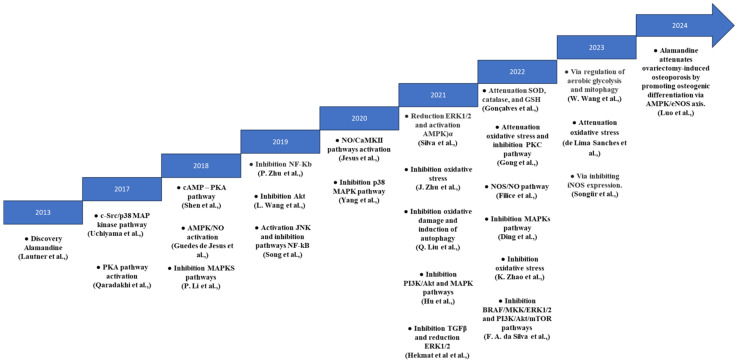
Chronological discoveries of Alamandine’s mechanisms of action [3,6,8,9,13,14,15,16,17,18,19,20,21,22,23,24,25,26,27,28,29,30,31,32,33,34].

**Figure 4 biomedicines-13-01957-f004:**
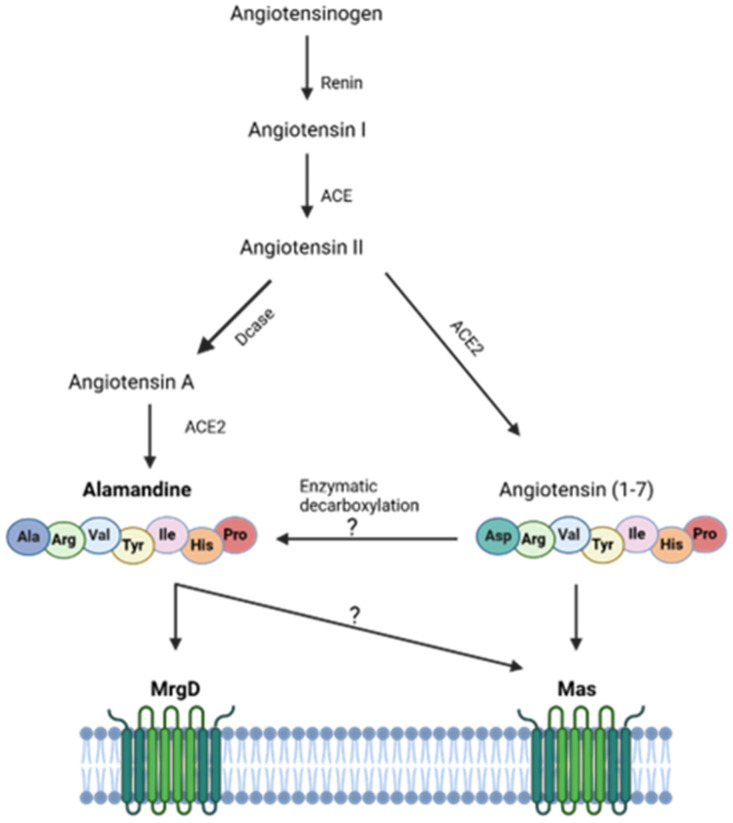
The classic renin–angiotensin system cascade. ACE: angiotensin-converting enzyme; ACE2: angiotensin-converting enzyme 2; Dcase: decarboxylase. Schematic representations were created with Biorender.com (YF28M19KKO).

**Figure 5 biomedicines-13-01957-f005:**
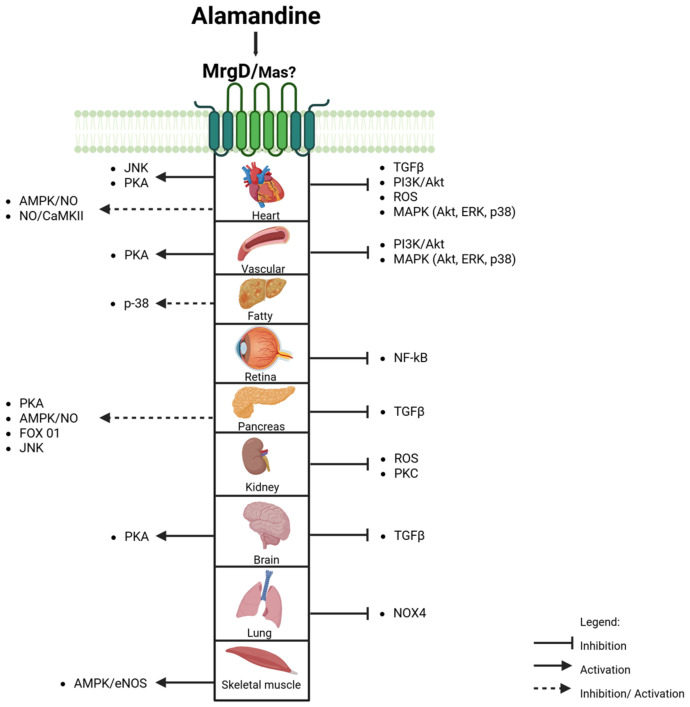
Organs and mechanisms activated by Alamandine. JNK—c-Jun N-terminal; PKA—protein kinase A; AMPK—adenosine monophosphate-activated protein kinase; NO—nitric oxide; TGFβ—transforming growth factor-β; PI3K—phosphoinositide 3-kinase; Akt—protein kinase B; ROS—Reactive oxygen species; MAPK—mitogen-activated protein kinase; ERK—extracellular signal-regulated kinase; NF-κB- nuclear factor kappa B; PKC—protein kinase C. Schematic representations were created with Biorender.com (CM28M18VE2).

## Data Availability

The raw data supporting the conclusions of this article will be made available by the authors upon request.

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
