# Peer review of "Biological Actions of Alamandine: A Scoping Review"

_biomedicines, 2025, doi:10.3390/biomedicines13081957_

Round 1
Reviewer 1 Report
Comments and Suggestions for Authors
The manuscript entitled “Biological actions of alamandine: a scoping review” provides a timely and informative overview of the current understanding of alamandine and its possible mechanisms of action. The figures and tables are of high quality and visually enhance the manuscript. However, several revisions are necessary to enhance the scientific depth and narrative coherence of the manuscript before it can be considered for publication.
1. Introduction
Please expand this section by including a brief overview of the two axes within the renin–angiotensin system (RAS). This addition would provide readers with a more comprehensive understanding of the physiological framework in which alamandine operates and better underscore how its discovery contributes to advancing the broader field of RAS research.
2. Results
1) Figure 2: Two suggestions to improve clarity and reader engagement:
a) Use distinct colors to highlight China and Brazil on the map, given that these countries contribute the majority of publications.
b) The current textual description of this figure reads more like a figure legend. Please provide a more detailed analysis in the results section, interpreting the geographic trends shown in the figure.
2) Figure 3 requires a clearer and more detailed explanation within the text.
3. Discussion
Given the limited number of recent publications on alamandine, a paragraph discussing the current research gaps and limitations in the field would significantly enhance this section. For example, this may include challenges related to the compound’s pharmacological targeting, mechanistic ambiguities, or translational research barriers.
4. Conclusion
Consider adding a paragraph that outlines future research directions, such as potential areas for preclinical or clinical investigation. Additionally, emphasize alamandine’s therapeutic potential, particularly in the context of cardiovascular, renal, or metabolic diseases, to reinforce the clinical significance of this molecule.
Author Response
We thank the reviewers for their constructive comments, which have greatly contributed to improving the clarity, depth, and scientific rigor of our manuscript. Below, we provide a point-by-point response to each comment, indicating the modifications made in the revised version of the manuscript.
Considering that the modifications were extensive, involving additions and removals of text, three files are being submitted:
- A response to the reviewers, briefly indicating acceptance and the line numbers where the requested changes were made;
- A clean version of the revised manuscript; and
- A detailed point-by-point response.
REVIEWER #1
The manuscript entitled “Biological actions of alamandine: a scoping review” provides a timely and informative overview of the current understanding of alamandine and its possible mechanisms of action. The figures and tables are of high quality and visually enhance the manuscript. However, several revisions are necessary to enhance the scientific depth and narrative coherence of the manuscript before it can be considered for publication.
COMMENT #1:
- Introduction
Please expand this section by including a brief overview of the two axes within the renin–angiotensin system (RAS). This addition would provide readers with a more comprehensive understanding of the physiological framework in which alamandine operates and better underscore how its discovery contributes to advancing the broader field of RAS research.
RESPONSE: Thank you for your valuable feedback. As requested, the introduction has been revised to include the relevant contextual information. The modified text has been incorporated at the beginning of the Introduction section, spanning lines 40 to 59 in the revised manuscript.
COMMENT #2:
- Results
1) Figure 2: Two suggestions to improve clarity and reader engagement:
- a) Use distinct colors to highlight China and Brazil on the map, given that these countries contribute the majority of publications.
RESPONSE: Figure 2 (line 144) has been updated accordingly: distinct colors were applied to highlight Brazil and China, the two countries with the highest number of publications on the topic. This modification was made to improve both clarity and reader engagement, as recommended.
COMMENT #3: b) The current textual description of this figure reads more like a figure legend. Please provide a more detailed analysis in the results section, interpreting the geographic trends shown in the figure.
RESPONSE: Thank you for your feedback on the figure description. We've included a more detailed interpretative text in the manuscript from lines 145 to 160. This section provides an in-depth analysis of the geographic trends illustrated in the figure, highlighting significant patterns and their implications.
COMMENT #4:
2) Figure 3 requires a clearer and more detailed explanation within the text.
RESPONSE: We've included a more detailed interpretative text in the manuscript from lines 161 to 171, highlighting significant patterns and their implications.
COMMENT #5:
- Discussion
Given the limited number of recent publications on alamandine, a paragraph discussing the current research gaps and limitations in the field would significantly enhance this section. For example, this may include challenges related to the compound’s pharmacological targeting, mechanistic ambiguities, or translational research barriers.
RESPONSE: Thank you for your valuable feedback regarding the need to discuss the research gaps and limitations in the field of alamandine. While we had included a paragraph addressing this topic, we fully agree that a more in-depth exploration would be advantageous. In response, we have expanded our discussion in lines 336 to 344 to provide a more comprehensive overview. This revised section now delves into several key areas, such as the complexities of pharmacological targeting, the mechanistic ambiguities surrounding alamandine's effects, and the translational research barriers that hinder the application of findings from basic research to clinical settings. We believe this enhanced focus will significantly enrich the manuscript and offer readers a clearer understanding of the challenges that lie ahead in alamandine research.
COMMENT #6:
- Conclusion
Consider adding a paragraph that outlines future research directions, such as potential areas for preclinical or clinical investigation. Additionally, emphasize alamandine’s therapeutic potential, particularly in the context of cardiovascular, renal, or metabolic diseases, to reinforce the clinical significance of this molecule.
RESPONSE: Thank you for this important suggestion. We have changed the Conclusion section (lines 346 to 361) to include future research directions, including the exploration of ALA in preclinical models of metabolic and cardiovascular diseases, and the need for early-phase clinical studies. We also emphasized its therapeutic potential based on its anti-inflammatory, antifibrotic, and cardioprotective properties.
Reviewer 2 Report
Comments and Suggestions for Authors The authors present a scoping review of the experimental data collected since 2013, when alamandine was discovered. The literature search was conducted using the JBI methodology. The sections Introduction, Materials and Methods, and Results are very well written. The discussion is comprehensive. My only comments are regarding the abbreviations and the presentation of cited authors in the text. In principle, if a term appears in the text 3 or more times, it could be presented as an abbreviation. Authors should thoroughly review all abbreviations, observing this rule and writing abbreviations correctly, including lowercase and uppercase letters (AMPK/ampk; PI3K/pi3k). For example line 51 - Mas receptor (MrgD) lines 196-197 - Protein Kinase B (PKB, also known as Akt) line 227 - Nitric Oxide Synthase and many others. Since the text is full of abbreviations, which makes it difficult to follow, where the abbreviation is for only one or two words, they can be written out, and no abbreviation should be given for them (for example, STMN1 - Stathmin 1, LF - Lung Fibroblasts, HK2 - Hexokinase 2). Regarding the abbreviations presented at the end of the article, they should present only the abbreviations that apply to the text of the article. Regarding supplemetary materials, at the end of the tables, give abbreviations that only apply to these additional tables. Regarding the names of the authors cited in the text, the numbers in the cited literature should also be given. For example, line 232 - Songür et al.(2023) and Hu et al.(2021) showed... and many others.Author Response
We thank the reviewers for their constructive comments, which have greatly contributed to improving the clarity, depth, and scientific rigor of our manuscript. Below, we provide a point-by-point response to each comment, indicating the modifications made in the revised version of the manuscript.
Considering that the modifications were extensive, involving additions and removals of text, three files are being submitted:
- A response to the reviewers, briefly indicating acceptance and the line numbers where the requested changes were made;
- A clean version of the revised manuscript; and
- A detailed point-by-point response.
REVIEWER #2
Comments and Suggestions for Authors
The authors present a scoping review of the experimental data collected since 2013, when alamandine was discovered. The literature search was conducted using the JBI methodology. The sections Introduction, Materials and Methods, and Results are very well written. The discussion is comprehensive.
COMMENT #1: My only comments are regarding the abbreviations and the presentation of cited authors in the text. In principle, if a term appears in the text 3 or more times, it could be presented as an abbreviation. Authors should thoroughly review all abbreviations, observing this rule and writing abbreviations correctly, including lowercase and uppercase letters (AMPK/ampk; PI3K/pi3k). For example line 51 - Mas receptor (MrgD) lines 196-197 - Protein Kinase B (PKB, also known as Akt) line 227 - Nitric Oxide Synthase and many others. Since the text is full of abbreviations, which makes it difficult to follow, where the abbreviation is for only one or two words, they can be written out, and no abbreviation should be given for them (for example, STMN1 - Stathmin 1, LF - Lung Fibroblasts, HK2 - Hexokinase 2).
RESPONSE: We appreciate the reviewer’s careful observation regarding the use of abbreviations. We have thoroughly reviewed the manuscript to ensure that abbreviations are introduced only when the terms appear three or more times, following the appropriate conventions for uppercase and lowercase letters (e.g., AMPK, PI3K).
Abbreviations used three times (or less) were deleted, except for CAMKII, which appears only in one paragraph. In this case, the abbreviation was maintained to reduce text length, and because its meaning was immediately accessible from the context, being mentioned just above. We have decided to maintain only the full descriptions for the following terms: Stathmin 1, c-Jun N-terminal kinase, Calmodulin-dependent kinase II, Ischemia-Reperfusion Injury, Malondialdehyde, Spontaneously Hypertensive Rats, and Lung Fibroblasts.
All instances such as "Mas receptor (MrgD), line 72; "Protein Kinase B (PKB, also known as Akt)", and "Nitric Oxide Synthase" have been revised accordingly.
Given that alterations were made throughout the entire text, it would be extremely tedious to present all the individual lines where changes were implemented. Instead, we are submitting a file with all the changes clearly visible throughout the text using the “track changes” tool.
COMMENT #2: Regarding the abbreviations presented at the end of the article, they should present only the abbreviations that apply to the text of the article.
RESPONSE: Thank you! Your keen observation helped us recognize areas needing improvement, and it has been instrumental in enhancing the clarity and consistency of our manuscript. We truly appreciate your attention to detail and are grateful for your contribution to refining our work. We have reviewed the list of abbreviations presented at the end of the article and ensured that only those abbreviations used in the main text remain in the list. Any abbreviations not used in the text were removed to maintain clarity and relevance. We are submitting a file with all the changes clearly visible throughout the text using the “track changes” tool.
COMMENT #3: Regarding supplemetary materials, at the end of the tables, give abbreviations that only apply to these additional tables.
RESPONSE: We have now included a separate abbreviations at the end of the supplementary tables section. These abbreviations apply exclusively to the supplementary content and do not overlap with those in the main manuscript, thus avoiding confusion.
COMMENT #4: Regarding the names of the authors cited in the text, the numbers in the cited literature should also be given. For example, line 232 - Songür et al.(2023) and Hu et al.(2021) showed... and many others.
RESPONSE: We acknowledge the importance of precise citation formatting. All citations in the text now include the corresponding reference numbers, following the journal’s style. For example, "Songür et al. (2023) [xx]" and "Hu et al. (2021) [xx]" have been updated throughout the manuscript to reflect this format consistently. We are submitting a file with all the changes clearly visible throughout the text using the “track changes” tool.